# Every View Counts: Cross-View Consistency in 3D Object Detection with Hybrid-Cylindrical-Spherical Voxelization

**Qi Chen**[*]
Johns Hopkins University
Baltimore, MD, USA

**Lin Sun**[*]
Samsung Semiconductor, Inc.
San Jose, CA, USA

**Ernest Cheung**
Samsung Semiconductor, Inc.
San Jose, CA, USA

**Alan Yuille**
Johns Hopkins University
Baltimore, MD, USA

## Abstract

Recent voxel-based 3D object detectors for autonomous vehicles learn point cloud representations either from bird eye view (BEV) or range view (RV, a.k.a. perspective view). However, each view has its own strengths and weaknesses. In this paper, we present a novel framework to unify and leverage the benefits from both BEV and RV. The widely-used cuboid-shaped voxels in Cartesian coordinate system only benefit BEV feature map. Therefore, to enable learning both BEV and RV feature maps, we introduce Hybrid-Cylindrical-Spherical voxelization. Our findings show that simply adding detection on another view as auxiliary supervision will lead to poor performance. We proposed a pair of cross-view transformers to transform the feature maps into the other view and introduce cross-view consistency loss on them. Comprehensive experiments on the challenging NuScenes Dataset validate the effectiveness of our proposed method which leverages joint optimization and complementary information on both views. Remarkably, our approach achieved mAP of $55.8\%$, outperforming all published approaches by at least $3\%$ in overall performance and up to $16.5\%$ in safety-crucial categories like cyclist.

## 1 Introduction

With a great surge of autonomous vehicles and accessibility of cheaper laser sensors, e.g. LiDAR, learning directly from 3D LiDAR point clouds has become increasingly popular. Among LiDAR-based 3D object detectors, a line of works [1, 2, 3, 4] borrow the success of convolutional neural networks on 2D images, and group the unordered, irregular and sparse point clouds into cuboid-shaped volumetric grids, i.e. voxels. 3D feature maps are memory-consuming, and therefore most of the recent works [4, 5, 6, 2, 7, 8] project the feature maps into 2D at different stages in their pipelines. When choosing 2D representations, it is important that objects in the input point cloud are still visible in the projected view. In autonomous driving scenarios, object do not overlap in the bird's-eye-view (BEV) and the size of the objects are consistent regardless of its distance from the ego-vehicle. Hence each object projected into BEV remains visible. Alternatively, RV projection suffers from occlusion and object size variation with respect to distance but it generates dense features. Both BEV and RV representation are suitable for 3D detection. State-of-the-art voxel based detectors [4, 5, 2, 7] detect objects based on features from either BEV or RV.

Features in each view has their strengths and weaknesses. From BEV, rigid objects are usually kept a distance from each other so it is easy to separate the objects. However, some important targets (e.g. traffic cones) are tiny when viewed from BEV, and thus are hard to detect. From RV, similar to 2D images, objects may be partially occluded and appear as difference sizes at different ranges, i.e., distances to the sensor. Furthermore, existing RV-based detectors [5] lose depth information during projection, making it hard to localize accurately.

**Main Contributions**   We present a novel Cross-view Consistent Network (CVCNet) which leverages the advantages of both range view (RV) and Bird's-eye-view (BEV) in 3D detection. We highlight two main contributions in this work. Firstly, to the best of our knowledge, we are the first work that introduces the concept of Cross-view Consistency to 3D detection task. We discover that the performance will degrade if we simply add detection on another view as an auxiliary supervisory signal. We posit that object appearances on two views are different and it's hard for the network to learn the latent correlation and extract common features from two views. Based on the observation that the correspondences between two views have similar properties to Hough Transform, we propose a pair of Hough-Transform-like cross-view transformers that explicitly incorporate the correlation between two views and enforce consistency on the transformed features. We have conducted ablation studies and in-depth discussions to show that such consistency is a key factor to benefit from joint learning in BEV and RV.

Secondly, we designed a new Voxel representation, Hybrid-Cylindrical-Spherical (HCS) Voxels, which enables us to extract features for both RV and BEV in a unified coordination system. In contrast, the commonly used cuboid-shaped voxels based on Cartesian coordinates provide benefits to feature learning on BEV. Driven by outstanding performance of shared models that are applied to extract common low-level features across different tasks, our model uses the shared 3D network and two light-weight branches to adapt into different views. Our HCS Voxels play an essential role in this design as it contains all the dimensions needed for projection to RV and BEV.

Extensive experiments on NuScenes dataset [9] demonstrate that CVCNet outperforms all the published approaches in overall average precision (mAP). In particular, our mAP on pedestrians, motorcyclist and cyclist are $83.0\%$, $61.8\%$, $38.8\%$, which is at least $2.9\%$, $10.3\%$, and $16.5\%$ better than existing published methods. These results signify substantial safety improvement when our algorithm is applied to autonomous vehicles.

## 2   Related Work

### 2.1   Voxelization for Point Clouds

To transform point clouds into image-like grid structures so that convolutional neural networks can be applied, several works group point clouds into volumetric grids. Commonly used volumetric grids are cuboid-shaped ones under Cartesian coordinate system. VoxNet [10] represents the cuboid-shaped voxels as occupancy grids: if there are no points in that voxel, the grid value is $0$, or $1$ otherwise. To avoid quantization effects of occupancy grids and extract richer voxel features, VoxelNet [1] samples a fix number of points within each voxel and applies Voxel Feature Extractor (VFE, a small PointNet [11] made of fully connected layers and a max pooling layer) to points in each voxel to extract voxel features. For efficiency, PointPillars [3] discretizes the 3D space into pillars so there is only one voxel along the height dimension.

Some recent works start to explore voxel shapes other than cuboids. Alsfasser et al [12] voxelizes points under the Cylindrical Coordinate System. PolarNet [13] groups points into 2D polar grids on BEV for semantic segmentation. MVF [14] adopts both cuboid-shape voxels and spherical voxels.

### 2.2   3D Detectors based on Single View

**3D Detection on BEV**   Detection on BEV is popular among voxel-based detectors. Approaches, e.g. PIXOR [4], based on 2D CNNs project point clouds into BEV. However, projection suffers from 3D structural information loss. To mitigate information loss, recent voxel-based detectors, such as VoxelNet [1], SECOND [2], Part$A^2$ [7] and Fast Point R-CNN [8], preserve the 3D structure during voxelization and adopt 3D CNNs at early or intermediate stages and finally project features to BEV and detect objects from BEV.

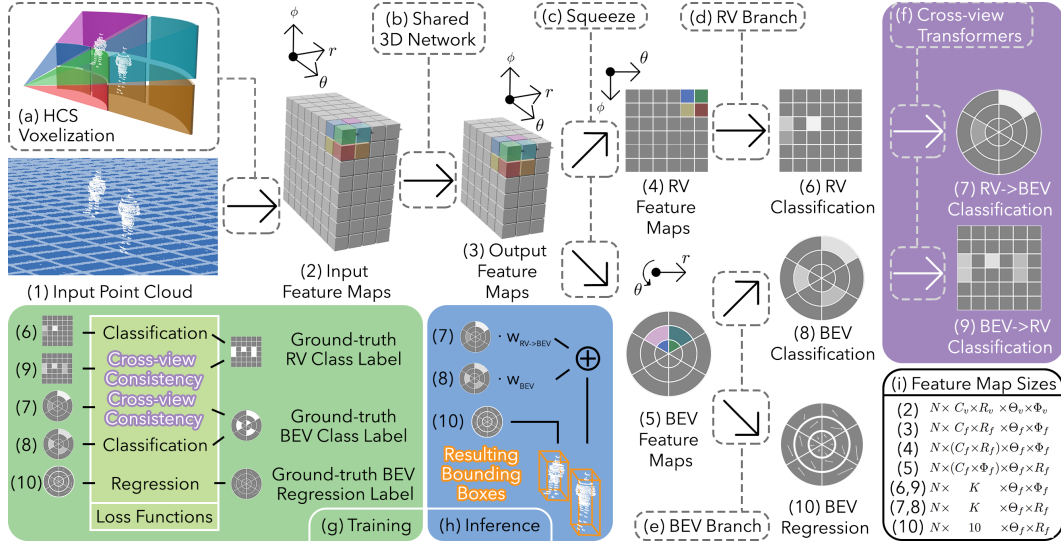

Figure 1: Overview of our approach: (a) Using HCS Voxelization, input point clouds are converted to voxel feature maps, and (b) passed into a 3D network shared between Ranged View (RV) and Bird's-Eye-View (BEV). The feature maps are then (c) squeezed and passed into (d) RV branch and (d) BEV branch with light-weight detection heads. (f) A pair of Cross-View Transformers align feature maps to alternative views. (g, h) In the green and blue box, we illustrate how we train our network and inference, respectively. (i) For the feature maps illustrated in this figure, we illustrate only the spatial dimensions, the full feature map sizes are shown in the white box. Note that the color of the voxels in (a, 2, 3, 4, 5) are consistent to illustrate our HCS voxels.

**3D Detection on RV**   There are very few works (LaserNet [5]) that learn representations from RV. RV is a compact representation that aligns with LiDAR scan pattern. But current RV detectors requires more data to perform well [5] and are outperformed BEV detectors on public datasets [15]. Occlusion and different scales of objects in RV also pose challenges to detection.

## 2.3   Other 3D Detectors

Point-based detectors generate proposals in 3D. PointRCNN [16], STD [17] and 3DSSD [18] generate proposals around segmented foreground points. Point-GNN [19] employs Graph Neural Networks as the feature extractor and proposals are generated around vertices. The complexity of point-based algorithms grows with the number of points so they have not gained popularity on datasets [9, 20] with a large number of points. MVF [14] uses a mixture of point and voxel representations. It fuses Cartesian and Spherical voxel features point-wisely by mapping them to the raw points and generate proposals around 3D points.

**Relation to MVF**   MVF also considers BEV and RV features in their pipeline but our approach is more efficient than MVF in following ways: 1) we voxelize points in a single shot thanks to HCS voxelization but MVF does it twice which consumes more time and memory; 2) in MVF, Cartesian voxels and Spherical voxels have different local contexts and thus it's basically a two-stream network that requires two separate 3D backbones to aggregate different local contexts for two views. Only point-wise features are shared. Ours has one sort of local context since we only have one type of voxels. This allows us to efficiently utilize shared 3D CNN to extract common low-level features for both views; 3) MVF finds cross-view correspondences by mapping voxels back to raw point clouds while we propose a neat solution - learnable cross-view transformers that densely align high level features on both views; 4) MVF simply do 3D detection on points, but we explicitly introduce supervision from RV and exploits the underlying cross-view consistency for joint optimization.

## 2.4   Multi-View Learning and Consistency

A stream of works incorporate multi-view inputs by aggregating the features. MV3D [6] and AVOD [21] fuse ROI features from point clouds and camera image for 3D object detection. For single-

Table 1: Different Voxelization Methods

| | Cartesian | Cylindrical | Spherical | HCS |
|---|---|---|---|---|
| 3D Voxel | $(x, y, z, \Delta x, \Delta y, \Delta z)$ | $(r, \theta, z, \Delta r, \Delta \theta, \Delta z)$ | $(R, \theta, \phi, \Delta R, \Delta \theta, \Delta \phi)$ | $(r, \theta, \phi, \Delta r, \Delta \theta, \Delta \phi)$ |
| BEV Voxel | $(x, y, \Delta x, \Delta y)$ | $(r, \theta, \Delta r, \Delta \theta)$ | N/A | $(r, \theta, \Delta r, \Delta \theta)$ |
| RV Voxel | N/A | N/A | $(\theta, \phi, \Delta \theta, \Delta \phi)$ | $(\theta, \phi, \Delta \theta, \Delta \phi)$ |
| Details | | $r = \sqrt{x^2 + y^2}$ $\theta = \arctan y/x$ | $R = \sqrt{x^2 + y^2 + z^2}$ $\theta = \arctan y/x$ $\phi = \arccos z/R$ | $r = \sqrt{x^2 + y^2}$ $\theta = \arctan y/x$ $\phi = \arctan z/r$ |
| References | VoxelNet [1], MVF [14] | Alsfasser et al [12] | MVF [14] | Ours |

modal inputs, MVCNN [22] utilizes a shared CNN to extract common features of object images rendered from different view angles. Consistency is widely used in multi-view geometry. Some depth estimation methods [23, 24, 25] rely on stereo image pairs and enforce photometric consistency. In multi-task learning, cross-task consistency such as the geometric constraints between depth, motion and optical flow [26] empirically improves generalization and stabilize model training.

# 3 Cross-View Consistent Network

The object detection problem is composed of two separate sub-problems: object classification, and bounding box regression. Therefore detectors often train two sets of feature maps to conduct classification to recognize objects, and bounding box regression to localize objects. According to the prior arts [5], detection on RV is difficult and requires more training data to achieve good performance. We also verify this on NuScenes dataset [9] where we observe that detection on RV is $16\%$ inferior to detection on BEV in overall mAP. Therefore, we design our network to take advantages from both BEV and RV to determine the object categories, but only use BEV feature maps to solve bounding box regression. Our overall approach is described in Figure 1.

## 3.1 Input Representation

The LiDAR point clouds consists of $N$ points, and each point is represented by a vector of point feature $f_p = (r_p, \theta_p, \phi_p, x_p, y_p, z_p, i_p, t_p)$, where $(x_p, y_p, z_p)$ is its location in Cartesian coordinates. $r_p$ is the range of the point on the horizontal plane. $\theta_p$, $\phi_p$ are the azimuth and elevation of the point observed from the LiDAR sensor. $i_p$ is the reflection intensity and $t_p$ is the timestamp when the LiDAR point is captured. Points are accumulated from maximum 10 successive frames in total to obtain denser point clouds. The points from previous frames are motion-compensated and transformed to the current frame.

## 3.2 Voxelization

The mathematical formulations of the common voxelization methods are presented in Table 1. The N/A entries in the table indicates that the original papers did not apply their work to the corresponding view. We design our proposed coordinate system by replacing $R = \sqrt{x^2 + y^2 + z^2}$ in Spherical coordinates system with $r = \sqrt{x^2 + y^2}$ which is adopted in Cylindrical coordinate system. This newly designed HCS system enables us to extract features for both RV and BEV in a unified coordination. The RV in the proposed coordinate system provides a wedge-shape frustum which can better handle the occlusion issues that commonly exist in the RV detection problem. Moreover, the wedge-shape voxel in HCS is more aligned with our perceptual system. Voxel partitions have fine grid in close range and coarse grid in the distance. The formulations of HCS in BEV and RV is shown in the table as well. We represent voxel features by randomly sampling $T$ points in each voxel and taking the mean of point features.

## 3.3 Shared 3D Backbone and Dual-View Branches

The low-level features are then extracted by a shared 3D Convolution Neural Network (CNN). We use the same 3D feature extractor as in CBGS [27], the start-of-the-art detector on NuScenes dataset [9]. Its architecture is made of sparse 3D convolution layers and similar to ResNet [28]. The output 3D feature map is squeezed to BEV and RV by merging $r$ dimension and $\phi$ dimension into channels of the feature maps, respectively.

BEV and RV branch are each composed of a 2D CNN. We adopt the same architecture for these two branches. It's a UNet-like architecture [29] which is widely adopted in recent state-of-the-art detectors [2, 27, 1]. Similar to CBGS [27], we add multi-group classification heads and bounding box regression heads to BEV branch and multi-group classification heads to RV branch. All the group heads are lightweight $1 \times 1$ convolutions. Additional details of the shared 3D CNN and the 2D CNN are provided in the supplementary materials.

### 3.4 Cross-View Transformers

When detecting the 3d objects in one scene, features from RV and BEV have the same semantic meaning, and therefore should be consistent across both views. To align the features from both views, rather than regarding detection on each view as independent supervisory signals, we design cross-view transformers to map RV features to BEV space and vice versa. We introduce new losses to enforce matching between the transformed features and target view labels, which we call cross-view consistency losses and is later defined in Section 4.2.

The size of classification confidence maps for BEV and RV are $K \times \Theta_f \times R_f$ and $K \times \Theta_f \times \Phi_f$, respectively, where $K$ is the number of categories excluding background, and $R_f, \Theta_f, \Phi_f$ are the dimensions for range $r$ on the horizontal plane, azimuth $\theta$ and elevation $\phi$. To align features from both views, we transform classification confidences from RV to BEV using $f_{RV \rightarrow BEV}$ and from BEV to RV by applying $f_{BEV \rightarrow RV}$. Interestingly, we find the correspondences between locations in BEV and RV have similar properties to Hough Transform: a location $(k_c, \theta_c, r_c)$, i.e. a point, on BEV corresponds to a column of locations $\{(k_c, \theta_c, \phi) | \phi = 1, 2, ..., \Phi_f\}$, i.e. a line segment, on RV, where $k_c, \theta_c, r_c$ denote constants. A location on RV also maps to a column of locations on BEV. This property is similar to one of the properties of Hough Transform, i.e. a point in one domain corresponds to a line in another. Inspired by Hough Transform, we propose a voting scheme to accumulate confidences on a column of locations on a source view to its corresponding location on target view. Taking $f_{RV \rightarrow BEV}$ as an example, the transform is a linear function:

$$f_{RV \rightarrow BEV}: \quad C^{BEV}_{(k_c, \theta_c, r_c)} = \sum_{\phi = 1, 2, ..., \Phi_f} w_{r_c, \theta_c, \phi} \times C^{RV}_{(k_c, \theta_c, \phi)} \tag{1}$$

where $C$ denotes the confidence score in each location and the weight $w_{r_c, \theta_c, \phi}$ can be either positive or negative which can be learned from a $1 \times 1$ convolution layer so the entire framework is fully convolutional.

## 4 Target Encoding and Joint Training

### 4.1 Target Encoding

We follow HotSpotNet [30] to assign targets. HotSpotNet adopts an anchor-free detection head that is flexible and can be easily adapted to feature maps with different voxel shapes. We briefly visit the target assignment policy below.

**Classification**   We assign locations in ground-truth bounding boxes as positive examples. To balance the number of positive examples in each ground-truth box ($b_k$) of different sizes, $M$ is set as the maximum number of positive examples in $b_k$, where $k$ denotes the category index. If there are more than $M$ positive examples inside $b_k$, only the top $M$ nearest locations of points to the bounding box center $(x_b, y_b, z_b)$ are chosen as postive examples. Denote this neighboring region as $\mathcal{N}_M(x_b, y_b, z_b)$. For a location $(i, j)$ on the feature map $\mathcal{F}_v$ ($v \in \{BEV, RV\}$), the target assignment policy is:

$$label_{(i,j)} = \begin{cases} k, & (i, j) \in b_k^v \ \& \ (i, j) \in \mathcal{N}_M(x_b, y_b, z_b) \\ 0, & (i, j) \notin \forall b_k^v \\ -1, & \text{else} \end{cases} \tag{2}$$

where $b_k^v$ is projected $b_k$ to view $v$. We set $M$ as a constant number. Labels with $-1$ will be ignored and do not contribute to the gradient descent. These refer to locations that are inside $b_k^{gt}$ but not in $\mathcal{N}_M(x_b, y_b, z_b)$ or inside more than one ground-truth bounding boxes. The label is encoded as a $K$-dimension one-hot vector.

**Bounding Box Regression**    We regress $B_{gt} = (d_x, d_y, z, \log l, \log w, \log h, \cos(rot), \sin(rot), v_x, v_y)$ as a 10-dimension vector, where $d_x, d_y$ are the deviation from the positive example to ground-truth bounding box center in Cartesian coordinates, $rot$ is bounding box orientation and $v_x, v_y$ are the velocities of target object along $x, y$ axis. HopSpotNet regresses $d_x, d_y, z$ as the outputs of soft $argmin$ to mitigate regression target scale imbalance. In order to allow our network to handle more categories, we further add $\log l, \log w, \log h$ as the outputs of soft $argmin$.

## 4.2   Joint Training

Denote $x$ as the input, $y$ as the ground-truth labels, $\alpha$, $\beta$, $\gamma$ and $\zeta$ to be the loss weights.

Loss on BEV is the weighted sum of classification loss and regression loss:

$$\mathcal{L}_{BEV} = \mathcal{D}(y_{BEV}, f_{BEV}) + \mathcal{D}(B_{gt}, \hat{B}) = \alpha\mathcal{L}_{cls}^{BEV} + \beta\mathcal{L}_{loc}^{BEV} \tag{3}$$

where $\mathcal{D}$ is the loss function calculator between the ground truth and the predictions, e.g. it can be the cross-entropy loss or focal loss for classification, smooth-L1 or L1 loss for bounding boxes regression. $\hat{B}$ is the predicted bounding boxes. Loss on RV only contains classification loss :

$$\mathcal{L}_{RV} = \mathcal{D}(y_{RV}, f_{RV}) = \gamma\mathcal{L}_{cls}^{RV} \tag{4}$$

**Cross-view consistency loss**    When using the Cross-View Transformers described in Section 3.4 to transform feature maps from a source view to its target view, we constrain them to be consistent with the labels in the target view. Therefore, we define the cross-view consistency losses as:

$$\mathcal{L}_{cvc} = \sum_{v,v' \in \{BEV,RV\}, v \neq v'} \mathcal{D}(y_v, f_{v' \to v} \circ f_{v'}(x)) \tag{5}$$

where $f_{v' \to v}$ denotes a cross-view transformer and $\circ$ is the function composition operator. Cross-View Consistency Loss is a classification loss where the predictions are the transformed confidence scores from source view $v'$ to target view $v$ and the targets are the labels on target view $v$, as below:

$$\mathcal{D}(y_v, f_{v' \to v} \circ f_{v'}(x)) = \zeta\mathcal{L}_{cls}^{v' \to v} \tag{6}$$

**Overall loss**    We apply Focal loss [31] to classification and Smooth-L1 loss [32] for bounding box regression. The final loss is the sum of losses on BEV, RV and cross-view consistency losses:

$$\mathcal{L} \triangleq \mathcal{L}_{BEV} + \mathcal{L}_{RV} + \mathcal{L}_{cvc} \tag{7}$$

## 4.3   Testing Phase

In testing time, final confidence score is obtained by blending BEV scores and RV→BEV scores, with weights $w_{BEV} = 0.8$ and $w_{R2B} = 0.2$ respectively.

## 5   Implementation

**Network Details**    $r$, $\theta$, and $\phi$ range is $[0.5, 51.1]$, $[-3.141, 3.141]$, and $[-1.3, 0.8]$ and the shell voxel size is $(0.1, 0.003, 0.0125)$. The max number of points per voxel is 8. We set loss weights $\alpha = \beta = \gamma = \zeta = 1$. Additional details can be found in the supplementary materials.

**Augmentation**    Class-balanced grouping and sampling is adopted as CBGS [27]. We conduct random flip in the $\theta$-axis, scaling with a scale factor sampled from [0.95, 1.05], rotation around $z$ axis between [-0.3925, 0.3925] rad and translation in range $[0.2, 0.2, 0.2]$ m in $x, y, z$ axis. To increase the ratio of positive examples in the training data, we adopt database sampling in SECOND [2]. We create a ground-truth database using ground-truth points in the annotated frames. During training we randomly drop half of points off gt database and filter gt boxes with less than 5 points inside.

# 6 Experiments and Results

## 6.1 Dataset and Evaluation Metrics

We evaluate our CVCNet on the NuScenes 3D detection dataset [9]. The dataset contains $1,000$ scenes, including $700$ scenes for training, $150$ scenes for validation and $150$ scenes for test. Each scene is of $20s$ duration and captured by 32-beam LiDAR. $40,000$ frames are annotated in total, including 10 object categories. The mean average precision (mAP) is based on the distance threshold (i.e. $0.5m, 1.0m, 2.0m$ and $4.0m$). Additionally, NuScenes detection score (NDS) [9], is introduced as a weighted sum of mAP and precision on box location, scale, orientation, velocity and attributes.

## 6.2 Comparison with state-of-the-art approaches

We submitted the results of our CVCNet to the NuScenes test server. In Table 2, we compare our test-set performance to state-of-the-art methods on the official leaderboard. Our submitted result is based on one single model without bells and whistles such as multi-scale testing used by CBGS [27]. Our approach achieves a significant improvement over the winner of 2019 NuScenes Detection challenge, CBGS[27], by a margin of $3\%$ in overall mAP. We argue the large gain mainly comes from joint optimization of detection on BEV and RV with cross-view consistency and complementary information from both views, which we show in following ablation studies.

Table 2: 3D detection mAP on the NuScenes test set

| Method | car | truck | bus | trailer | constr-uction vehicle | pede-strian | motor-cycle | bike | traffic cone | barr-ier | mAP | NDS |
|---|---|---|---|---|---|---|---|---|---|---|---|---|
| SARPNET [33] | 59.9 | 18.7 | 19.4 | 18.0 | 11.6 | 69.4 | 29.8 | 14.2 | 44.6 | 38.3 | 31.6 | 49.7 |
| PointPillars [3] | 68.4 | 23.0 | 28.2 | 23.4 | 4.1 | 59.7 | 27.4 | 1.1 | 30.8 | 38.9 | 30.5 | 45.3 |
| WYSIWYG [34] | 79.1 | 30.4 | 46.6 | 40.1 | 7.1 | 65.0 | 18.2 | 0.1 | 28.8 | 34.7 | 35.0 | 41.9 |
| CBGS [27] | 81.1 | 48.5 | 54.9 | 42.9 | 10.5 | 80.1 | 51.5 | 22.3 | **70.9** | 65.7 | 52.8 | 63.3 |
| CVCNet (Ours) | **82.6** | **49.5** | **59.4** | **51.1** | **16.2** | **83.0** | **61.8** | **38.8** | 69.7 | **69.7** | **55.8** | **64.2** |

## 6.3 Ablation Studies

Table 3: Ablation study of how training with different losses affect inference from a single view. 3D detection mAP is evaluated on the NuScenes validation set. B: loss on BEV; R: loss on RV; B→R: consistency loss from BEV to RV; R→B: consistency loss from BEV to RV.

| | Supervision | | | | Inference | | car | truck | bus | trailer | constr-uction vehicle | pede-strian | motor-cycle | bike | traffic cone | barr-ier | mAP |
|---|---|---|---|---|---|---|---|---|---|---|---|---|---|---|---|---|---|
| | B | R | B→R | R→B | BEV | RV | | | | | | | | | | | |
| a) | ✓ | | | | ✓ | | 82.1 | 47.2 | 56.7 | 30.9 | 13.5 | 80.0 | 50.4 | 27.6 | 56.8 | 64.9 | 51.0 |
| b) | ✓ | ✓ | | | ✓ | | 82.1 | 46.9 | 59.7 | 30.9 | 11.0 | 79.8 | 44.7 | 24.6 | 53.6 | 64.7 | 49.8 |
| c) | ✓ | ✓ | ✓ | | ✓ | | 82.7 | 47.7 | 59.4 | 31.1 | 14.0 | 80.4 | 52.7 | 32.7 | 60.5 | 64.6 | 52.6 |
| d) | ✓ | ✓ | | ✓ | ✓ | | 82.8 | 48.6 | 61.6 | 32.6 | 16.2 | 80.7 | 54.7 | 30.9 | 59.7 | 65.8 | 53.4 |
| e) | ✓ | ✓ | ✓ | ✓ | ✓ | | 82.8 | 49.1 | 61.8 | 33.4 | 19.0 | 80.8 | 53.3 | 33.0 | 60.2 | 65.4 | 53.9 |
| f) | ✓ | ✓ | | ✓ | | ✓ | 79.1 | 39.1 | 52.7 | 23.2 | 16.9 | 77.9 | 44.7 | 25.8 | 55.8 | 61.7 | 47.7 |
| g) | ✓ | ✓ | ✓ | ✓ | | ✓ | 79.1 | 39.5 | 52.0 | 23.2 | 14.7 | 78.2 | 47.7 | 28.2 | 56.5 | 60.3 | 48.0 |
| h) | ✓ | ✓ | ✓ | ✓ | ✓ | ✓ | 83.2 | 50.0 | 62.0 | 34.5 | 20.2 | 81.2 | 54.4 | 33.9 | 61.1 | 65.5 | 54.6 |

**Consistency Losses for Joint Optimization** We evaluate detection results models trained with different supervisions in Table 3. **a) & b)**: when detection from BEV only is evaluated, by simply adding loss on RV as auxiliary supervisory signal, mAP drops from $51\%$ to $49.8\%$. The object appearances on BEV and RV are quite different so it's hard for the network to learn the correlation between two views and hence the common representations for both views are not effectively learned. **c), d) & e)**: when the correlation between BEV and RV is explicitly incorporated into the network by cross-view transformers and consistency losses, detection results from BEV are improved. And RV→BEV consistency boosts the performance on BEV most significantly. By adding consistency losses, we improve overall mAP for detection on BEV from $51\%$ to $53.9\%$. **f) & g)**: we also see that consistency also improves detection on RV, showing consistency helps detection on BEV and RV respectively by joint optimization. We implement detection from RV by transforming RV

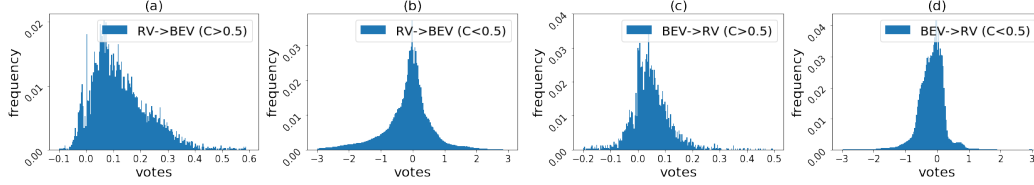

Figure 2: Histogram of votes from transformers. A vote denotes contribution from a location on source view $v'$ to its corresponding location to target view $v$. $f_{RV \rightarrow BEV}$: (a) votes from RV locations where confidences $C > 0.5$; (b) votes from RV locations where $C < 0.5$. $f_{BEV \rightarrow RV}$: (c) votes from BEV locations where confidences $C > 0.5$; (d) votes from BEV locations where confidences $C < 0.5$.

classification confidences to BEV (RV→BEV) and combining them with bounding box regression results from BEV branch. Detection on RV is possible only when transformer from RV to BEV is applied and thus consistency loss from RV to BEV is applied too. Note that as we explained in Section 3, we abandon regression on RV due to its poor performance.

**Complementary information on different categories**   To further analyze how incorporating information from RV and consistency losses affect each category on BEV, we find that the performance gain is associated with the area of the object on BEV and RV. We only observe slight performance gain (0.7%) on cars. The BEV area for cars is $9.1m^2$ on average while RV area is $8.8m^2$. The area on RV is smaller than on BEV so incorporating RV does not make cars easier to detect. However, we observe significant improvement on motorcycles, bicycles and traffic cones. Taking traffic cones as an example, the BEV area is $0.18m^2$, which makes it really hard to detect on BEV. The area on RV is $0.63m^2$, adding more exposure of this category and increasing its chances to be detected.

**Blending scores from both views**   In Table 3 e), g) & h) we show by blending classification confidences from both view together during inference, i.e. classification confidences from BEV and RV→BEV, we achieve better overall mAP (54.6%) than inference from a single view (53.9% and 48.0% respectively).

## 6.4  Understanding Cross-View Transformers

In Section 3.4, we present cross-view transformers inspired by Hough Transform with learnable weights implemented by $1 \times 1$ conv. To understand what the transformers are really doing, we demonstrate the statistical distribution of votes from transformers $f_{RV \rightarrow BEV}$ and $f_{BEV \rightarrow RV}$. Taking $f_{RV \rightarrow BEV}$ as an example, it transforms classification confidences from each location on RV to their corresponding locations on BEV. A vote denotes contribution from a location $(k_c, \theta_c, \phi)$ on RV to its corresponding location $(k_c, \theta_c, r_c)$ on BEV, i.e. $w_{r_c, \theta_c, \phi} \times C^{RV}_{(k_c, \theta_c, \phi)}$ in Equation 1. The final transformed classification confidence, $C^{BEV}(k_c, \theta_c, r_c)$, is the sum of votes across $\{(k_c, \theta_c, \phi) | \phi = 1, 2, ..., \Phi_f\}$, a column of locations on RV. In Figure 2 (a), at RV locations where confidences are high ($C > 0.5$) for the presence of an object, we observe most locations give positive votes to their corresponding locations on BEV. This shows that when the network is very certain that there is an object on RV, it will vote for its presence on BEV. Interestingly, Figure 2 (a) shows the votes follow Gumbel distribution. However, when the network is unsure whether there is an object on RV ($C < 0.5$), the votes can be either positive or negative, and the chances for the votes to be positive or negative are approximately equal. Figure 2 (b) shows the votes where the confidences are low follow Laplace distribution. We observe similar behavior from $f_{BEV \rightarrow RV}$ (Figure 2 (c)(d)). This is consistent with our intuition that high likelihood of object appearance on view $v'$ also means high likelihood of object appearance on its corresponding location on view $v$, but a location on view $v$ corresponds to a column of locations on view $v'$ and it must reason all locations across that column on view $v'$.

**Discussions**   We also tried other Hough-Transform-like functions such as average pooling and max pooling with broadcasting but the mAP drops by 44% and 4% respectively, compared to our learnable transformers. Average pooling can be regarded as equal votes on each location along a column of locations. But the column contains both positive and negative examples. Average pooling does not discriminate different locations along the column, and therefore performs extremely bad. Though

max pooling is inferior to our learnable transformers, it does not hurt the performance compared to evaluating on BEV only (Table 3b). Max pooling only considers the most confident locations, which points to the right direction as the presence of object in one view indicates its possible presence on the corresponding locations on the other view. In that sense, our learnable transformer can be regarded a soft version of max pooling, but also reasons other locations along the column.

## 6.5 Comparisons with MVF

In order to further verify the effectiveness of CVCNet, we designed the following experiment to make a fair comparison with MVF [14]. The experiments are conducted on the Waymo Open Dataset which is a large-scale dataset recently released for benchmarking object detection algorithms at industrial production level and the performance is shown in Table 4. The value ranges we used in this experiment for $r$, $\theta$, and $\phi$ are $[0.5, 76]$, $[-3.141, 3.141]$, and $[-0.41, 0.84]$ respectively, and the shell voxel size is $(0.2, 0.003, 0.003)$. We applied the same augmentation methods used in NuScenes experiments during the training. Our approach outperforms all one stage detectors in overall mAP for vehicle detection.

Table 4: Vehicle detection mAP for one-stage detectors on Waymo validation set

| Method | LEVEL 1 3D IoU=0.7 | | | |
| --- | --- | --- | --- | --- |
| | Overall | 0-30m | 30-50m | 50m-$\infty$ |
| StarNet [35] | 53.70 | - | - | - |
| PointPillars | 56.62 | 81.01 | 51.75 | 27.94 |
| PPBA [36]+ PointPillars | 62.44 | - | - | - |
| MVF | 62.93 | 86.30 | 60.02 | 36.02 |
| AFDet [37] | 63.69 | **87.38** | 62.19 | 29.27 |
| CVCNet (ours) | **65.20** | 86.80 | **63.84** | **36.65** |

The performance can potentially be improved significantly by a thorough exploration of alternative parameters for the Waymo Open Dataset. In the results we show here, substantial improvements are observed in comparing to MVF, which is the only method that considers multiple views in the 3D detection framework.

Our algorithm runs at 11 FPS with a single V100 GPU on Waymo Open Dataset. Since MVF did not release the experimental details and code, it is difficult to make apple-to-apple comparisons with it on the inference speed and the number of parameters. However, given that MVF uses separate backbones, we also adopt our method to separate backbones for BEV and RV and present the results in Table 5, to highlight the advantages of our shared backbone approach in terms of inference speed and number of parameters. The experiments are conducted on NuScenes validation set. With comparable performances, adding one more backbone will incur an extra 240 ms of runtime (more than $267\%$) per frame and 30 MB of parameters (more than $18\%$) in our proposed CVCNet.

Table 5: Performance with separated or shared backbones on NuScenes val set

| Backbone | FPS | #params | car | truck | bus | trailer | constr-uction vehicle | pede-strian | motor-cycle | bike | traffic cone | barr-ier | mAP |
| --- | --- | --- | --- | --- | --- | --- | --- | --- | --- | --- | --- | --- | --- |
| separate | 3 | 201MB | 83.1 | **50.2** | 59.2 | 33.7 | 16.0 | 81.0 | **57.1** | **34.6** | 60.9 | **66.7** | 54.2 |
| shared | 11 | 171MB | **83.2** | 50.0 | **62.0** | **34.5** | **20.2** | **81.2** | 54.4 | 33.9 | **61.1** | 65.5 | **54.6** |

## 7 Conclusions

We propose a novel framework, Cross-view Consistent Network (CVCNet), to leverage the benefits from BEV and RV. In contrast to existing 3D LiDAR-based detectors that use Cartesian voxelization, we propose Hybrid-Cylindrical-Spherical voxelization, which enables learning from both BEV and RV in one network. We present a dual-view architecture and formulate detection on both views as a multi-view learning problem. Instead of simply treating detection on BEV and RV as separate supervisions, we introduce cross-view transformers and enforce cross-view consistency on both views. Experimental results on the NuScenes and Waymo Open Dataset demonstrate that our approach significantly improves overall detection accuracy, and therefore enhances safety of autonomous vehicles. We hope that our work will further enlighten the multi-view or multi-sensor fusion tasks.

## Broader Impact

3D detection is the first stage in the computational pipeline for a self-driving car. Just as perception enables humans to make instant associations and act on them, the ability to identify what and where the visual targets are from immediate surroundings is a fundamental pillar for the safe operation of an autonomous vehicle. The pandemic of COVID-19 manifests greater needs for autonomous driving and delivery robots, when contact-less delivery is encouraged. Though there is controversy about the ethics for autonomous vehicles especially for their decision making, robust 3D detection with higher accuracy is always desired to improve safety. In addition, LiDAR point clouds do not capture person identity and thus 3D detection on LiDAR point clouds does not involves privacy issue.

## Acknowledgement

We thank Dr. X.Y.Z. for useful discussions that greatly improved the manuscript.

## Footnotes

* indicates equal contributions

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
