[Supplementary Material]

# Supplementary Materials for "Every View Counts: Cross-View Consistency in 3D Object Detection with Hybrid-Cylindrical-Spherical Voxelization"

**Qi Chen**\*
Johns Hopkins University
Baltimore, MD, USA

**Lin Sun**\*
Samsung Semiconductor, Inc.
San Jose, CA, USA

**Ernest Cheung**
Samsung Semiconductor, Inc.
San Jose, CA, USA

**Alan Yuille**
Johns Hopkins University
Baltimore, MD, USA

In this document we provide more details about implementation and experiments about different voxelization methods.

## 1 Implementation

### 1.1 Network Details

The model architecture is shown in Figure 1.

### 1.2 Training Details

**Parameters for Detection Heads** To regress $d_x$, $d_y$, we set 15 bins and range $[-4, 4]$ for soft $argmin$. For $z$ we set 12 bins and range $[-5, 3]$. We set 12 bins and range $[-3, 3]$ for $\log l, \log, w, \log h$. We use code weight 0.5 for velocity prediction and 1.0 for other bounding box statistics. In the classification head, we set alpha=0.25, gamma=2.0 for Focal Loss [1].

**Miscellaneous** We use adamW [2] optimizer together with one-cycle policy [3] with LR max 0.04, division factor 10, momentum ranges from 0.95 to 0.85, fixed weight decay 0.01 to achieve super convergence. With batch size 5, the model is trained for 20 epochs. During inference, top 1000 proposals are kept in each group, and NMS with score threshold 0.1 is applied. Max number of boxes allowed in each group after NMS is 80.

## 2 Comparing different voxelization

We compare the effects of different voxelization in Table 1. For fast implementation, we use PointPillars [4] as backbone here. We train our models with different voxelization methods only on BEV. By using approximately the same number of voxels, Cartesian Voxelization and HCS voxelization achieve comparable mAP. We also notice more even distribution of points in voxels as we can see the average number of points per voxel and the standard deviation is smaller by HCS voxelization because it places more voxels where points are dense and fewer voxels where points are sparse. Thus HCS voxelization increases voxel utilization by reducing the number of empty voxels.

Figure 1: Network Architecture. Left: 3D Shared CNN. Right: BEV/RV Branch. Each block consists of Convolution, BatchNorm, ReLU. The values in each block represent kernel sizes and the number of output feature channels respectively. For BEV branch, $C = 128$. For RV branch, $C = 64$.
.

Table 1: 3D detection mAP on the NuScenes val set: comparing Cartesian voxelization with HCS voxelization

| voxelization | #voxels | #non-empty voxels | voxel utilization | voxel size | #points per voxel | mAP |
|---|---|---|---|---|---|---|
| Cartesian | $254K$ | $19K$ | $7.6\%$ | $0.2m \times 0.2m$ | $9 \pm 11.3$ | $42.0$ |
| HCS | $251K$ | $53K$ | $20.9\%$ | $0.25m \times 0.005rad$ | $4.7 \pm 6.1$ | $42.4$ |
| Cartesian | $160K$ | $14K$ | $8.9\%$ | $0.25m \times 0.25m$ | $10.8 \pm 12.6$ | $40.0$ |
| HCS | $157K$ | $36K$ | $22.7\%$ | $0.25m \times 0.008rad$ | $6.4 \pm 7.6$ | $39.8$ |

## Footnotes

\* indicates equal contributions