[Reviews · NeurIPS 2020]

Review 1

Summary and Contributions: The main contribution of the paper is a method that leverages cross-view consistency between BEV and RV point cloud representation by using Hybrid-Cylindrical-Spherical voxelization. The performance is good as demonstrated on NuScenes dataset.

Strengths: 1. The idea of combining the advantage of both BEV and RV point cloud representation is great. And the paper proposes an effective way to combining features from both view. 2. The performance on NuScenes is good. I like the ablation study which clearly shows the contributions of each component. 3. The whole paper is well written. The motivation and the design is clearly described.

Weaknesses: 1. Experiments are only conducted on NuScenes. It would be great if the KITTI / Waymo dataset is also evaluated. 2. The method is compared to a few baselines, but some popular methods such as AVOD, F-PointNet, PointRCNN are not listed. 3. The runtime of inference is not reported, which is important for practical application.

Correctness: yes

Clarity: yes

Relation to Prior Work: yes

Reproducibility: Yes

Additional Feedback: I think the overall quality of the paper is good. The proposed method for combining two different view of point cloud representation is elegent and reasonable. And the authors addressed my major concern by adding results on Waymo dataset, which are pretty good compared to existing methods. Therefore I decide to keep my rating as acceptance.


Review 2

Summary and Contributions: This paper proposes a Cross-view Consistent Network (CVCNet) to leverage the benefits from BEV and RV. It proposes Hybrid-Cylindrical-Spherical voxelization that enables learning from both BEV and RV in one network. It proposed a pair of cross-view transformers to transform the feature maps into the other view and introduce a cross-view consistency loss on them as a multi-view learning problem.

Strengths: It proposed a novel Cross-view Consistent Network (CVCNet) to leverage the advantages of both range view (RV) and Bird’s-eye-view (BEV) in 3D detection. + introduces the concept of Cross-view Consistency to 3D detection task. + proposes a pair of Hough-Transform-like cross-view transformers that explicitly incorporate the correlation between two views and enforce consistency on the transformed features. + designed a new Voxel Representation, Hybrid-Cylindrical-Spherical (HCS) Voxels, which enables to extract features for both RV and BEV. + experiments on NuScenes dataset show substantial improvement over state of art methods

Weaknesses: The main baseline for this paper is the MVF paper. Despite the claim in the beginning section one paragraph details the potential advantage of the proposed method over the MVF paper including better memory and time efficiency and better utilization of context, it does not back it up with any experiments. It would be more convincing if the paper do a side by side comparison with MVF on common datasets such as Kitti and show significant improvement. The paper however is trained on another dataset not the ones MVF tested. This is the main point I didn't give the paper a higher score. Clearly MVF[32] is the main reference paper and could be the paper inspired the current work. The proposed paper try to propose some better ways, which are all legit, however it is very disappointing the experiments does not include a side by side comparison on the two open datasets Waymo and Kitti. Even for the Nuscene dataset it didn't include MVF 32 in the table. I checked the supplementary it is not there either.

Correctness: seems okay

Clarity: Okay.

Relation to Prior Work: yes

Reproducibility: Yes

Additional Feedback: I suggest try more than the dataset [2], do side by side comparison with MVF 32 both in accuracy, memory, parameters, timing, etc. That would make a more convincing case. After rebuttal. Based on the rebuttal I increased my original rating.


Review 3

Summary and Contributions: This paper proposes a method for detecting objects in LIDAR data. The idea is to consider both a "range view" (ie from the LIDAR position) and a "bird eye view" (ie a top view) of the Lidar data for the input of 2 CNNs (1 per view). A term that constrains the output of the 2 CNNs to be consistent is then introduced to the loss function used for training the 2 CNNs (introducing constraints between output has been shown to improve performance on other problems). This term works by computing linear combinations of output terms for one view along lines in 3D. Each linear combination should be consistent with 1 output term for the other view. The weights of the linear combinations are also optimized during learning. This is close to a recent method (MVF [32]). This is discussed in the paper (section 2.3). I agree the method proposed here is more elegant than [32]

Strengths: Adding terms constraining different outputs has been done before (for example for constraining depth and normal predictions), but the proposed solution is designed for a different and important problem. The method is simple (which is a good thing). The experiments seem to confirm the validity of the approach.

Weaknesses: The main idea is specific to the problem considered by the paper - object detection in LIDAR data, but it is an important problem, so I think this part is fine. My main concern about the paper is the clarity of the text.

Correctness: The method is simple and well justified. The experiments are limited to a dataset (NuScenes) but this is sufficient to validate the method with an ablation study. The method is compared to very recent papers (CVPR 20, ..), with an improvement of ~3% of mAP compared to the best competitor method.

Clarity: That's my main concern about the paper. There are many small language mistakes, mostly in the technical section (Section 3), but they are not the main problem. The proposed method is simple (which, again, is something good), but somehow it is difficult to understand from the text. I try to detail below what could be changed to improve the text clarity: - Calling "Cross-view transformers" the mapping functions used in the constraint term is confusing, as "transformer" means other thing in deep learning (transformers in NLP, spatial transformers) - Section 3.4 (about the transformers) mentions features, while in fact it is the final outputs that are "transformed" - it is not said explicitly that the weights in Eq (1) are learned in Section 3.4 - Eqs (3) to (6) seem to use the Euclidean(?) norm, while the authors probably meant some similarity functions; - Eqs (6) is disconnected from the text - Figure 1 is very dense and it is difficult to understand the method from it, while it should be possible to convey visually the method in a simple way - mentioning the Hough transform to explain the method did not make the presentation more intuitive for me. The method is probably more related to epipolar constraints (but this is a detail, maybe a matter of taste)

Relation to Prior Work: yes. The Related Work section is pretty clear.

Reproducibility: Yes

Additional Feedback: Update after rebuttal and discussion: I still think the paper can be written in a much better way, but since I am the only one to have a problem with this and the authors promised to improve the writing, I increased my "overall score". My advice to the authors is to make a genuine effort to improve the readability - it would help the paper to get a stronger impact. I understand better the link with Hough Transform now, but alternatively the operation in Eq (1) can also be seen as a smart pooling that takes into account the 3D geometry of the problem. --------- My main motivation for the final score is the text quality. Given the number of papers published at NeurIPS, papers should be as easy as possible to understand. In principle, the problems in the text could be fixed, but they are many and it is difficult to expect the authors can fix all of them for the final version, which does not have a second reviewing stage.


Review 4

Summary and Contributions: They propose a novel Cross-view Consistent Network to leverage both RV and BEV for 3D detection task and design a new Hybrid-Cylindrical-Spherical Voxel Representation to extract features for both RV and BEV. Their proposed CVCNet outperforms all the published approaches in the metric of mAP on NuScenes dataset.

Strengths: The main idea is very novel and interesting. They first use cross-view features to detect 3D objects, and propose an interesting cross-view voxel presentations and cross-view network architectures. Performance on NuScenes dataset suggests improvements comparing with previous methods.

Weaknesses: No weaknesses I found

Correctness: Yes

Clarity: Yes

Relation to Prior Work: Yes

Reproducibility: No

Additional Feedback:

[Author Response · NeurIPS 2020]

We thank all the reviewers for their efforts and thoughtful feedbacks. All reviewers agreed that we proposed a novel and interesting Cross-view Consistency Network (CVCNet) to leverage the advantages of both range view (RV) and Bird's-eye-view (BEV) in 3D detection. In the paper, we presented the state-of-the-art performance on the 3D detection benchmark. Moreover, we provided in-depth analysis and clear ablation studies to validate our contributions. We will address the issues raised by reviewers in the final version.

**Experiments on extra datasets and comparisons (R1, R2)** - NuScenes and Waymo are relatively new and large 3D detection datasets (50x larger than KITTI w.r.t #scenes). A lot of previous state-of-the-art algorithms, such as AVOD and PointRCNN, did not evaluate their performance on NuScenes and Waymo. In the main paper, we already include all the published results on NuScenes. Additionally, we conduct the experiments on Waymo and the performance is shown in Table 1. Our approach outperforms all one stage detectors by a large margin in overall mAP for vehicle detection.

Table 1: Vehicle Detection mAP for One-stage Detectors on Waymo OD Validation Set

| Method | LEVEL 1 3D IoU=0.7 | | | |
| | Overall | 0-30m | 30-50m | 50m-∞ |
| --- | --- | --- | --- | --- |
| StarNet (3) | 53.70 | - | - | - |
| PointPillars | 56.62 | 81.01 | 51.75 | 27.94 |
| PPBA (1)+ PointPillars | 62.44 | - | - | - |
| MVF | 62.93 | 86.30 | 60.02 | 36.02 |
| AFDet (2) | 63.69 | 87.38 | 62.19 | 29.27 |
| CVCNet (ours) | **68.43** | **87.55** | **63.53** | **42** |

Our algorithm runs at 8 FPS with a single V100 GPU on Waymo Open Dataset. MVF reported 15 FPS but they did not specify the machine they used or if they optimized to speed up. Since MVF did not release the experimental details and code, it is difficult to make comparisons with it on inference speed and number of parameters. Since MVF uses separate backbones, to show some insights on speed and number of parameters, we present our method with separate backbones for BEV and RV in Table 2. The experiments are conducted on NuScenes validation set. With comparable performances, adding one more backbone will add extra 240 ms runtime (267% more) per frame and 30 MB of paramters (18% more) in our proposed CVCNet.

Table 2: Performance with Separate or Shared Backbones on NuScenes Val Set

| Backbone | FPS | #parameters | car | truck | bus | trailer | constr-uction vehicle | pede-strian | motor-cycle | bike | traffic cone | barr-ier | mAP |
| --- | --- | --- | --- | --- | --- | --- | --- | --- | --- | --- | --- | --- | --- |
| separate | 3 | 201MB | 83.1 | **50.2** | 59.2 | 33.7 | 16.0 | 81.0 | **57.1** | **34.6** | 60.9 | **66.7** | 54.2 |
| shared | 11 | 171MB | **83.2** | 50.0 | **62.0** | **34.5** | **20.2** | **81.2** | 54.4 | 33.9 | **61.1** | 65.5 | **54.6** |

**Text clarity (R3)** - Thank you for your comments about text clarity. We will carefully revise our paper in the final version. We will make all the equations, as well as the figure in the paper clear and easy to understand. Meanwhile, we should say the other reviewers didn't mention any problems with the writing.

Sorry we don't fully understand your comment "calling cross-view transformers the mapping functions used in the constraint term is confusing". Did you mean L161 "map RV features to BEV space"? We did not write it as a constraint term. This is how we match voxels between two views.

We do see some researchers call output scores features (4) but we will clarify this in the final version.

With our Hybrid-Cylindrical-Spherical Voxelization, a voxel in one view corresponds to a column of voxels in the other view. This property is similar to one of the properties of Hough Transform, ie. a point in one domain correponds to a line in another domain. Our transformers are inspired by voting in Hough Transform. We understand it may not be easy to grasp this property without visualizations. We are considering making dynamic figures for people to get a better idea of it. Epipolar geometry is related in the sense of multi-view correspondence. However, epipolar geometry does not apply to our work because it assumes pinhole camera model while BEV is from orthographic projection.

[1] Cheng, S., Leng, Z., Cubuk, E.D., Zoph, B., Bai, C., Ngiam, J., Song, Y., Caine, B., Vasudevan, V., Li, C., et al.: Improving 3d object detection through progressive population based augmentation. arXiv preprint arXiv:2004.00831 (2020)

[2] Ge, R., Ding, Z., Hu, Y., Wang, Y., Chen, S., Huang, L., Li, Y.: Afdet: Anchor free one stage 3d object detection. arXiv preprint arXiv:2006.12671 (2020)

[3] Ngiam, J., Caine, B., Han, W., Yang, B., Chai, Y., Sun, P., Zhou, Y., Yi, X., Alsharif, O., Nguyen, P., et al.: Starnet: Targeted computation for object detection in point clouds. arXiv preprint arXiv:1908.11069 (2019)

[4] Vora, S., Lang, A.H., Helou, B., Beijbom, O.: Pointpainting: Sequential fusion for 3d object detection. In: Proceedings of the IEEE/CVF Conference on Computer Vision and Pattern Recognition. pp. 4604–4612 (2020)


[Meta-Review · NeurIPS 2020]

The paper proposes a method for LIDAR-based object detection that exploits cross-view consistency between bird's-eye view and range view point clouds of the scene. The two inputs are fed to separate neural networks trained with a loss function that includes a term that encourages consistency between the two representations. Evaluations demonstrate strong performance compared to baselines on NuScenes. The paper was reviewed by four knowledgeable referees, who read the author response and subsequently discussed the paper. The reviewers agree that the manner in which the method exploits the bird's-eye and range views is interesting and elegant, namely the HCS voxel representation that enables feature extraction for both views and the manner in which the method enforces consistency on the transformed feature representations. Experimental results on NuScenes show the method's promise, while the ablations help to convey the contributions of the different model components. The reviewers raised important questions about the experimental evaluation. Some of these questions were addressed in the author response, including the additional evaluation on the Waymo dataset. However, there are issues with the way that the paper is currently written and the authors are strongly urged to address these in the next version of the paper.